# Post Quantum Cryptographic Keys Generated with Physical Unclonable Functions

**Bertrand Cambou \*, Michael Gowanlock**  **, Bahattin Yildiz, Dina Ghanaimiandoab, Kaitlyn Lee, Stefan Nelson, Christopher Philabaum, Alyssa Stenberg and Jordan Wright**

College of Engineering Informatics and Applied Sciences (CEIAS), Northern Arizona University (NAU), Flagstaff, AZ 86011, USA; michael.gowanlock@nau.edu (M.G.); bahattin.yildiz@nau.edu (B.Y.); dg856@nau.edu (D.G.); kdl222@nau.edu (K.L.); swn34@nau.edu (S.N.); cp723@nau.edu (C.P.); ajs937@nau.edu (A.S.); jaw566@nau.edu (J.W.)

\* Correspondence: bertrand.cambou@nau.edu; Tel.: +1-928-523-7824

**Featured Application: Using physical unclonable functions (PUFs), in support of networks secured with a public key infrastructure, to generate, on demand, the key pairs needed for lattice and code PQC algorithms.**

**Abstract:** Lattice and code cryptography can replace existing schemes such as elliptic curve cryptography because of their resistance to quantum computers. In support of public key infrastructures, the distribution, validation and storage of the cryptographic keys is then more complex for handling longer keys. This paper describes practical ways to generate keys from physical unclonable functions, for both lattice and code-based cryptography. Handshakes between client devices containing the physical unclonable functions (PUFs) and a server are used to select sets of addressable positions in the PUFs, from which streams of bits called seeds are generated on demand. The public and private cryptographic key pairs are computed from these seeds together with additional streams of random numbers. The method allows the server to independently validate the public key generated by the PUF, and act as a certificate authority in the network. Technologies such as high performance computing, and graphic processing units can further enhance security by preventing attackers from making this independent validation when only equipped with less powerful computers.

**Keywords:** lattice cryptography; code cryptography; post quantum cryptography; physical unclonable function; public key infrastructure; high performance computing

## 1. Introduction

In most public key infrastructure (PKI) schemes for applications such as cryptographic currencies, financial transactions, secure mail and wireless communications, the public keys are generated by private keys with Rivest–Shamir–Adleman (RSA) and elliptic curve cryptography (ECC). These private keys are natural numbers, typically 3000-bit long for RSA and 256-bits long for ECC. For example, in the case of ECC, the primitive element of the elliptic curve cyclic group is multiplied by the private key to find the public key. It is now anticipated that quantum computers (QC) will be able to break both RSA and ECC when the technology to manufacture enough quantum nodes becomes available. The paper entitled "A Riddle Wrapped in an Enigma" by N. Koblitz and A. J. Menezes suggested that the ban of RSA and ECC by the National Security Agency is unavoidable, and that the risk of QC is only one element of the problem [1]. Plans to develop post quantum cryptographic (PQC) schemes have been proposed to secure blockchains by Kiktenko et al. [2], and for cryptocurrency security by Semmouni et al. [3], even if the timeline for the availability of powerful QC is highly speculative. Recently, Campbell et al. [4], and Kampanakisy et al. [5], proposed distributed ledger cryptography and digital signatures with PQC. In 2015, the National Institute of Standards and Technology (NIST) initiated a large-scale program to

standardize PQC algorithms. One possible implementation of PQC algorithms for a PKI is the one in which each client device, or designate, generates the key pairs, and sends the public keys to a certificate authority (CA). This assumes a separate authentication process, and that each client device can securely store the key pairs.

The research question that is the subject of this paper is the feasibility of using physical unclonable functions (PUFs), together with a handshake process with the CA that generate new key pairs from the PUF at each transaction, thereby eliminating the need to store the key pair. Attempts to retrieve the secret keys are not relevant anymore as they are only used once. Such a configuration is raising several structural and technical questions. A secure enrollment process of each PUF needs to be established, and the CA has to store the challenges and reference values of each PUF. Such an infrastructure is already known when the security is based on secure hardware elements and tokens and requires special protections against opponents. From a technical standpoint, it is questionable that the long key pairs necessary for PQC algorithms can be generated from physical elements. While a single bit mismatch is not acceptable for PQC algorithms, the natural drifts of PUFs over environmental conditions and aging are real concerns that need to be addressed. This paper is structured in the following way:

[**Section** 2]: The lattice and code-based cryptographic algorithms under consideration for standardization by NIST are presented. These algorithms are well documented, and the software stack written in C can be downloaded for IoT implementation. The schemes are based on the generation of random numbers, and the computation of public–private key pairs. The digital signature algorithms (DSA), and key encapsulation mechanisms (KEM) are not more complex to implement with PQC than with the existing asymmetrical cryptographic schemes.

[**Section** 3]: We present some of the challenges associated with the use of PUF technology to secure PKI architectures. The proposed methods are based on existing cryptographic schemes, and commercially available PUFs. We present how the response based cryptographic (RBC) scheme can overcome the bit error rates (BER) that occur when keys are generated from physical elements. Finally, we present some hardware considerations in the implementation of PQC for PKI.

[**Section** 4]: In this section, we propose schemes that use PUFs to generate the public–private key pairs for lattice and code-based cryptography. We show how the combination of random number generators, combined with the streams generated by the PUF can generate key pairs with relatively low error rates. We show how the error in these streams can be corrected using a search engine.

[**Section** 5]: Finally, in the implementation and experimental section, we compare cryptographic schemes and algorithms. We analyze the experimental results comparing the efficiency of RBC operating with various PQC schemes, ECC, and advanced encryption standard (AES). As expected, asymmetrical schemes are slower than AES; however, the performance of the selected PQC algorithms is encouraging for the implementation of PUF-based architecture, using the RBC to handle the expected BER.

## 2. Lattice and Code-Based Post Quantum Cryptography

In 2019, the number of candidates of the NIST standardization effort was narrowed to 26, as part of phase two of the program [6]. In July 2020, NIST announced the selection of seven likely finalists for phase three of the program [7]: CRYSTALS-Kyber, CRYSTALS-Dilithium, SABER, NTRU, and FALCON with lattice cryptography [8–12]; RAINBOW with multivariate cryptography [13], and Classic McEliece with code-based cryptography [14,15]. The software developed is mainly targeting DSA applications, as well as KEM. Lattice cryptography is relatively mature, well documented, and is most likely to become mainstream for cybersecurity. Lattice-based algorithms exploit hardness to resolve problems such as the closest vector problem (CVP), learning with error (LWE), and learning with rounding (LWR) algorithms, and share similarities with the knapsack cryptographic problem.

### 2.1. Learning with Error Cryptography

The LWE of the CVP problem was first introduced by Regev [16]. The knowledge of integer-based vector $t$, and matrix $A$ with $t = A.s_1$ cannot hide the vector $s_1$; however, the addition of a "small" vector of error $s_2$ with $t = A.s_1 + s_2$, makes it hard to distinguish the vectors $s_1$ and $s_2$ from $t$. The vector $s_2$ needs to be small enough for the encryption/decryption cycles, but large enough to prevent a third party from uncovering the private key ($s_1$; $s_2$) from the public information ($t$; $A$). The public–private cryptographic key pair generation for client device i can be based on polynomial computations in a lattice ring, and is described in Figure 1:

1. The generation of a first data stream called seed $a_{(i)}$ that is used for the key generation; in the case of LWE, the seed $a_{(i)}$ is shared openly in the network.
2. The generation of a second data stream called seed $b_{(i)}$ that is used to compute a second data stream for the private key $Sk_{(i)}$; the seed $b_{(i)}$ is kept secret.
3. The public key $Pk_{(i)}$ is computed from both data streams and is openly shared.
4. The matrix $A_{(i)}$ is generated from seed $a_{(i)}$.
5. The two vectors $s_{1(i)}$ and $s_{2(i)}$ are generated from seed $b_{(i)}$.
6. The vector $t_{(i)}$ is computed: $t_{(i)} \leftarrow A_{(i)} s_{1(i)} + s_{2(i)}$.
7. Both seed $a_{(i)}$ and $t_{(i)}$ become the public key $Pk_{(i)}$.
8. Both $s_{1(i)}$ and $s_{2(i)}$ become the private key $Sk_{(i)}$.

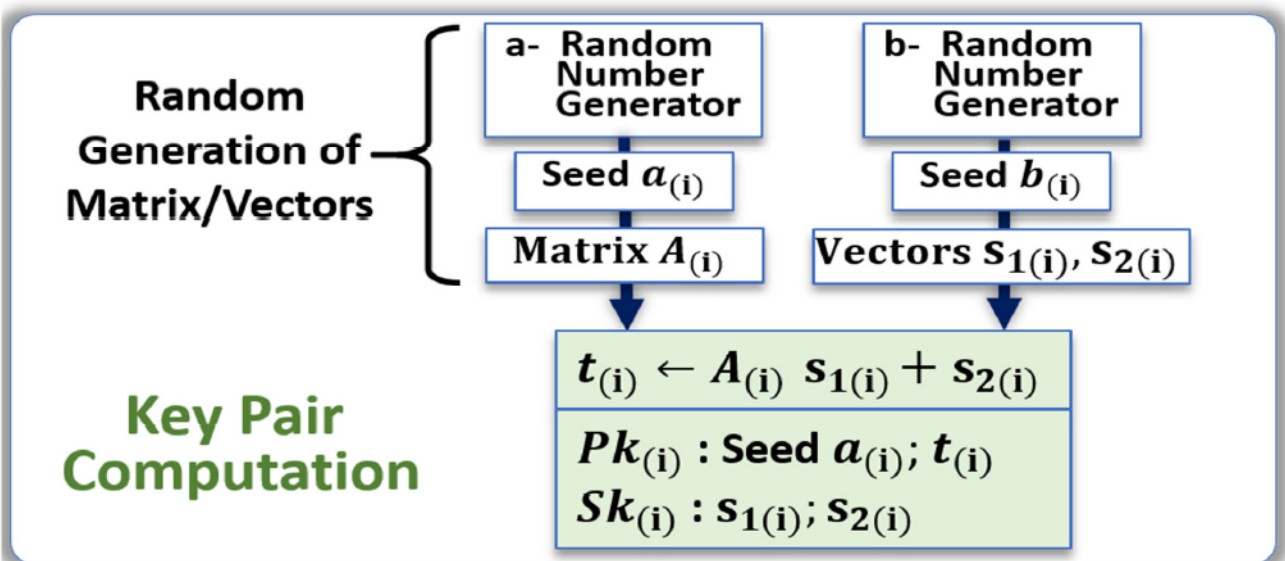

**Figure 1.** Example of public–private key generation for LWE based cryptography. The matrix $A_{(i)}$ and vectors $S_{1(i)}$ and $S_{2(i)}$ are generated from random number generators. The vector $t_{(i)}$ is computed from these elements for the generation of the key pair.

A digital signature algorithm (DSA) can be realized from the LWE instance by first generating a public–private key pair as in Figure 1. The secret key is then used to sign a message, and the public key is used to verify this signed message. In CRYSTALS-Dilithium [8], the authors use a Fiat-Shamir with Aborts approach [17] for their signing and verification procedure. The outline of the signing procedure is as follows:

1. Generate a masking vector of polynomials $y$.
2. Compute vector $A.y$ and set $w_1$ to be the high-order bits of the coefficients in this vector.
3. Create the challenge $c$, as the hash of the message and $w_1$.
4. Compute intermediate signature $z = y + c.s_1$.
5. Set parameter $\beta$ to be the maximum coefficient of $c.s_1$.
6. If any coefficient of $z$ is larger than $\gamma_1 - \beta$, then reject and restart at step 1.

7. If any coefficient of the low-order bits of $A.z - c.t$ is greater than $\gamma_2 - \beta$, then reject and restart at step 1.

Note: $\gamma_1$, $\gamma_2$, and $\beta$ are set such that the expected number of repetitions is between 4 and 7.

The general outline of the verification procedure is given by the following:

Compute $w_1'$ to be the high-order bits of $A.z - c.t$ and accept if all coefficients of $z$ are less than $\gamma_1 - \beta$ and if $c$ is the hash of the message and $w_1'$.

Encapsulation allows for two parties to securely share a symmetric key by encapsulating the key in ciphertext. When both parties have the symmetric key, they are then able to use a symmetric-key encryption algorithm to communicate (e.g., AES). These algorithms are known as key encapsulation mechanisms (KEM) and a few examples from NIST are SABER [18], Classic McEliece [14,15], CRYSTALS-Kyber [19], and NTRU [20]. The process of using encapsulation with LWE/LWR is described below:

- The public and private keys of both parties are constructed as described in Figure 1.
- Person A sends Person B their public key.
- Person B randomly generates a symmetric key and encapsulates it in a ciphertext with the public key of person A.
- Person B sends the ciphertext to person A.
- Person A decapsulates the ciphertext with their private key.
- Both parties now have the symmetric key in their possession.

In summary, LWE schemes are now relatively mature, and very well documented. The methods selected by the NIST standardization program, presented here, are straightforward to use. The codes are widely available online for download, and we successfully deployed them in our research environment to study the use of PUFs for key generation.

### 2.2. Learning with Rounding Cryptography

The learning with rounding problem was first introduced by Banerjee [21]. It is the derandomized version of learning with error, which deterministically generates the noise in the LWE by rounding coefficients. This will eliminate the noise sampling, and significantly reduce the bandwidth [22]. The LWR is proved to be as hard as LWE to solve; hence, it remains secure to be used in cryptographic applications. In schemes such as "Saber", a constant h is added as a constant vector to simulate the rounding operation by bit shifting, therefore playing a similar protecting role than the error vectors of LWE [18]. Saber, which is one of the NIST's finalists in the key encapsulation category, uses LWR for key generation in public key encryption and key encapsulation. Below all three steps of PKE and KEM are described:

Saber PKE Key Generation

1. Similar to LWE, seed $a_{(i)}$ is used to generate matrix $A_{(i)}$.
2. Seed $b_{(i)}$ is used to generate vector $s_{(i)}$.
3. The vector $t_{(i)}$ is computed: $t_{(i)} \leftarrow A_{(i)}. s_{(i)} + h_{(i)}$.
4. Both seed $a_{(i)}$ and $t_{(i)}$ become the public key $Pk_{(i)}$.
5. $s_{(i)}$ becomes the private key $Sk_{(i)}$.

Saber PKE Encryption

1. The seed $a_{(i)}$ and $t_{(i)}$ is extracted from public key to encrypt the message $m$.
2. Matrix $A_{(i)}$ and vector $s'_{(i)}$ are generated.
3. The vector $t'_{(i)}$ is computed by rounding the product of $A_{(i)}. s'_{(i)}$: $t'_{(i)} \leftarrow A'_{(i)}. s'_{(i)} + h_{(i)}$.
4. Polynomial $v'_{(i)}$ is calculated as: $v'_{(i)} = t_{(i)}. s'_{(i)}$.
5. $v'_{(i)}$ is used to encrypt the message $m$ which denoted as $c_m$.
6. Ciphertext consists of $c_m$ and $t'_{(i)}$.

Saber PKE Decryption

1. $v_{(i)}$ is calculated as: $v_{(i)} = t'_{(i)}. s_{(i)}$.

2. The message $m'$ is decrypted by reversing computations with $v_{(i)}$ and $c_m$.

The Saber key encapsulation mechanism has three steps: Saber KEM Key Generation, Saber KEM Encapsulation, and Saber KEM Decapsulation:

Saber KEM Key Generation

1. Saber PKE key generation is used to return seed $a_{(i)}$, $t_{(i)}$ and $s_{(i)}$.
2. Both seed $a_{(i)}$ and $t_{(i)}$ become the Saber KEM public key $Pk_{(i)}$.
3. The hashed public key $Pkh_{(i)}$ is generated using SHA3-256.
4. Parameter $z$ is randomly sampled.
5. $z$, $Pkh_{(i)}$ and $s_{(i)}$ become the Saber KEM secret key.

Saber KEM Encapsulation

1. Message $m$ and public key $Pk_{(i)}$ are hashed using SHA3-256.
2. Saber PKE encryption is used to generate ciphertext.
3. Hash of the $Pk_{(i)}$ and ciphertext are concatenated, then hashed to encapsulate the key.

Saber KEM Decapsulation

1. Message $m'$ is decrypted by using Saber PKE Decryption.
2. The decrypted message $m'$ and hashed public key $Pkh_{(i)}$ are hashed to generate K'.
3. Ciphertext $c'_m$ is generated from saber PKE Encryption for message $m'$.
4. If $c_m = c'_m$ then the K = Hash(K',c), if not, K = Hash(z,c).

The level of documentation available on LWR is not quite as complete as what is available for LWE. However, the proposed implementation of LWR for NIST's PQC program is solid. The use of PUFs to secure PKIs based on LWR is not more challenging than the one based on LWE.

### 2.3. NTRU Cryptography

Cryptographic algorithms such as FALCON, which uses NTRU (*Nth* degree of TRUncated polynomial ring) arithmetic, are also based on lattice cryptography. The parameters of the scheme include a large prime number N, a large number **q** and a small number **p** that are both used for modulo arithmetic. Two numbers **df** and **dg** are used to truncate the polynomials $f_{(i)}$ and $g_{(i)}$. The key generation cycle for client device (i), as shown in Figure 2, is the following:

1. Generation of the two truncated polynomials $f_{(i)}$ and $g_{(i)}$ from seed $a_{(i)}$ and seed $b_{(i)}$.
2. Computation of $Fq_{(i)}$, which is the inverse of polynomial $f_{(i)}$ modulo **q**.
3. Computation of $Fp_{(i)}$, which is the inverse of polynomial $f_{(i)}$ modulo **p**.
4. Computation of polynomial $h_{(i)}$: $h_{(i)} \leftarrow$ p. $Fq_{(i)}$. $g_{(i)}$.
5. The private key $Sk_{(i)}$ is $\{f_{(i)}; Fp_{(i)}\}$.
6. The public key $Pk_{(i)}$ is $h_{(i)}$.

As the polynomials $f_{(i)}$ and $g_{(i)}$ are not always usable, they are subject to some preconditions such as invertible modulo $p$ and $q$. The client device needs to try several possible random numbers, and select the ones giving acceptable private keys. Once sufficient public and private keys are available, the encryption of the plaintext message $m$, $m \in \{-1, 0, 1\}^N$ is done by finding the random polynomial $r$, $r \in \{-1, 0, 1\}^N$, which uses a corresponding parameter $d_r$, and calculating the ciphertext with the equation $e \equiv r.h + m \pmod{q}$. To retrieve $m$ from $e$, we first calculate $a \equiv f.e \pmod{q}$ and lift the coefficients of $a$ to be between $\pm q/2$. Then, $a \pmod{p}$ is equal to m. [23].

NTRU lattices can also be applied to DSA. This was originally introduced in NTRUSign, but NIST submissions such as Falcon expand on these algorithms [9]. Falcon utilizes the GPV framework applied to NTRU lattices; that is, the public key is a long basis for an NTRU lattice while the private key is a short basis. From here, the message m is sent a non-lattice point $C$, utilizing a random value salt and hash function $H$. Using the short basis, a user signs by finding the closest vector $v$ to $c$. The signature is (salt, $s = c - v$), verified by checking if $s$ is short and $H$ (msg ‖ salt) $- s$ is a point on the lattice (verified using the long basis [24]).

The cryptography based on NTRU is well known, and extremely well documented. The polynomial arithmetic truncating the N-th element is elegant and effective. Like other lattice algorithms under consideration by NIST, we are considering the NTRU as a strong candidate, both for DSA, and KEM.

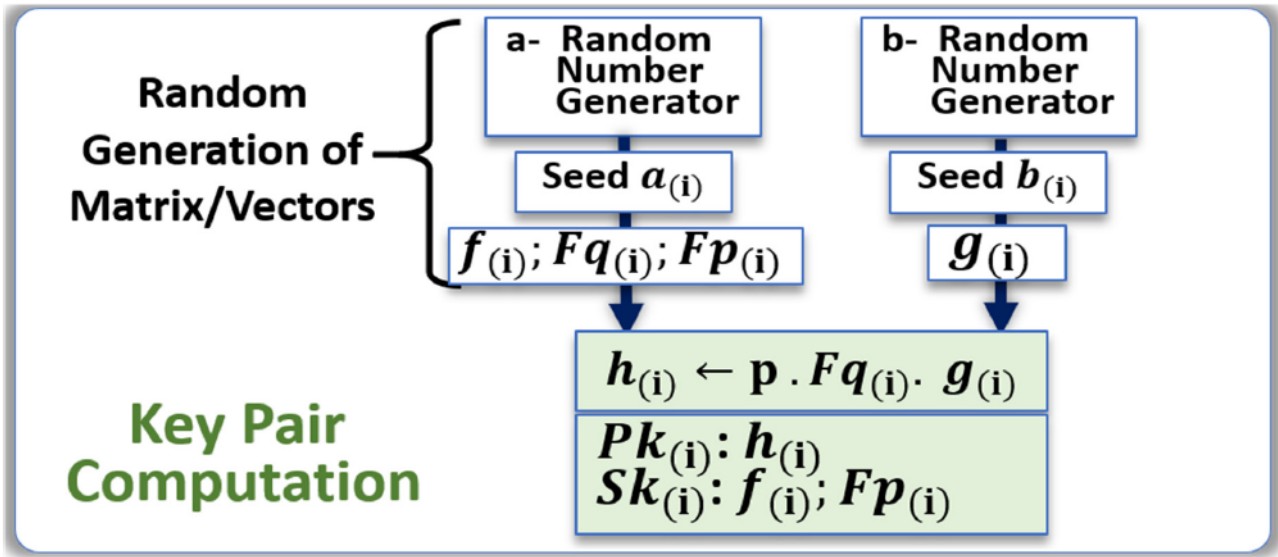

**Figure 2.** Example of key generation for NTRU cryptography. The polynomials $f_{(i)}$ and $g_{(i)}$ and are generated from random number generators, from which the inverses $Fq_{(i)}$ and $Fp_{(i)}$ are computed. The public key $h(i)$ is also computed from these polynomials.

### 2.4. Code-Based Cryptography

Code-based algorithms such as Classic McEliece are implemented with binary Goppa codes, that is, Goppa codes with underlying computations in finite Galois fields **GF(2$^{\mathbf{m}}$)**. The parameters are an irreducible polynomial of degree **t**, field exponent **m**, and code length **n**. The resulting code has error-correction capability of **t** errors, the information-containing part of the code word has a size of **k = n − m × t** and has generator matrix **G** with a size of **k × n** [14,15].

The block diagram of Figure 3 is showing an example of public–private key generation for code-based cryptography, and client device i.

1.  Seed $a_{(i)}$ is used to create a random invertible binary **k × k** scrambling matrix $S_{(i)}$.
2.  Seed $b_{(i)}$ is used to create a random **n × n** permutation matrix $P_{(i)}$.
3.  The public key $Pk_{(i)} = \hat{G}_{(i)}$ is computed with the generator matrix $G$: $\hat{G}_{(i)} \leftarrow S_{(i)}. \, G. \, P_{(i)}$
4.  The private key $Sk_{(i)}$ is $\{G; S_{(i)}{}^{-1}, P_{(i)}{}^{-1}\}$.

Given a generator matrix of a binary Goppa code $G$, an irreducible polynomial of degree $t$, the field exponent $m$, and the code length $n$, the encryption process involves the following steps:

1.  Create the public key, $\hat{G}_{(i)}$ as described above.
2.  Multiply the message $m$ by $\hat{G}_{(i)}$, creating the ciphertext message $\hat{m}$.
3.  Add a random error vector $e$ of Hamming weight $t$ to $\hat{m}$ to obtain the ciphertext **c.**

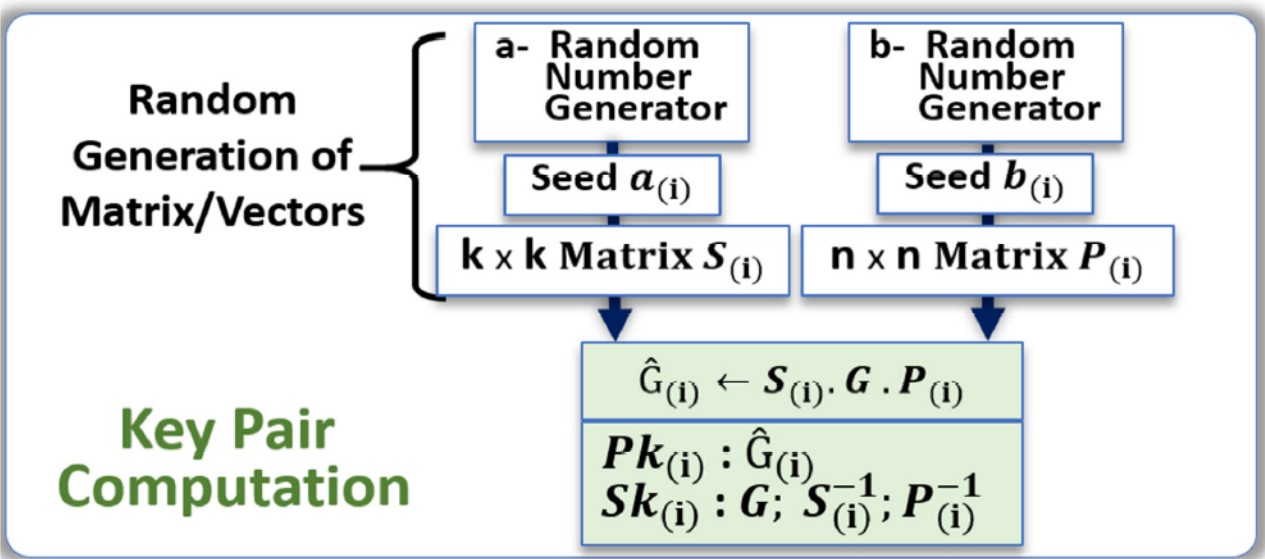

**Figure 3.** Example of key generation for code-based cryptography. The two matrixes $S_{(i)}$ and $P_{(i)}$ are generated from random number generators. The key pairs are computed from the matrixes, and the generator matrix G.

Given a ciphertext $c$, a decoding algorithm, and the private key $\{G; S_{(i)}^{-1}, P_{(i)}^{-1}\}$, decryption involves the following steps:

1. Compute $\hat{c} = c\, P_{(i)}^{-1}$.
2. Use the decoding algorithm to correct the errors to obtain $\hat{m}$.
3. Obtain the original message by computing $m = \hat{m}.S_{(i)}^{-1}$.

One example of a decoding algorithm is Patterson's algorithm. This algorithm calculates the error-locator polynomial which has roots corresponding with the locations of the error bits added to the encrypted message. This algorithm can be implemented as follows [25]:

**Input:** Syndrome polynomial $s$, Goppa polynomial g of degree $t$

**Patterson (s, g):**

1. $t = s^{-1} \bmod g$.
2. $t = \sqrt{t + x}$.
3. Find polynomials $a$, $b$ such that $b.t \equiv \bmod g$ with deg(a) $\leq$ $|t/2|$ and deg(b) $\leq$ $|(t\text{-}1)/2|$ using the extended Euclidean algorithm.
4. Calculate and return the error locator polynomial, $e = a^2 + x.b^2$.

Once the error locator polynomial is found, the Berlekamp Trace Algorithm can be used to find the roots of the polynomial via factorization. These correspond to the locations of the error bits added to the message. The Berlekamp Trace Algorithm can be implemented as follows [26]:

**Input:** Polynomial to factor $p$, trace polynomial $t$, basis index $i$

**Berlekamp Trace (p, t, i):**

1. if deg($p$) $\leq 1$
2. return the root of $p$.
3. $p_0 = gcd(p, t(B_i.\,x))$.
4. $p_1 = gcd(p, 1 + t(B_i.\,x))$.

return berlekampTrace ($p_0$, i + 1), berlekampTrace ($p_1$, i + 1).

Code-based is probably the most mature, and well documented PQC algorithm currently under consideration. The new implementations are highly effective for KEM; the extended output functions allow the quick generation of the two matrixes needed for key generation.

## 3. Public Key Infrastructure

### 3.1. Public-Private Key Pairs

As part of a PKI, the public–private key pairs can be used to securely transmit shared secret keys though KEM and to digitally sign messages with DSA (see Figure 4). The public key $Pk_{(2)}$ of Client 2 encapsulates the shared secret key of Client 1, that can only be viewed by the client (2), thanks to their private key $Sk_{(2)}$ that reverses the encapsulation. Client 1 uses their private key $Sk_{(1)}$ to digitally sign a message that is verified with the public key $Pk_{(1)}$, providing non-alteration and non-repudiation in the transaction. The trust and integrity of such an architecture relies on the following:

i.      The secure generation and distribution of the public–private key pairs to the client devices that are participating in the PKI.
ii.     The identification of the client devices, and trust in their public keys.
iii.    The sharing of the public keys among participants.

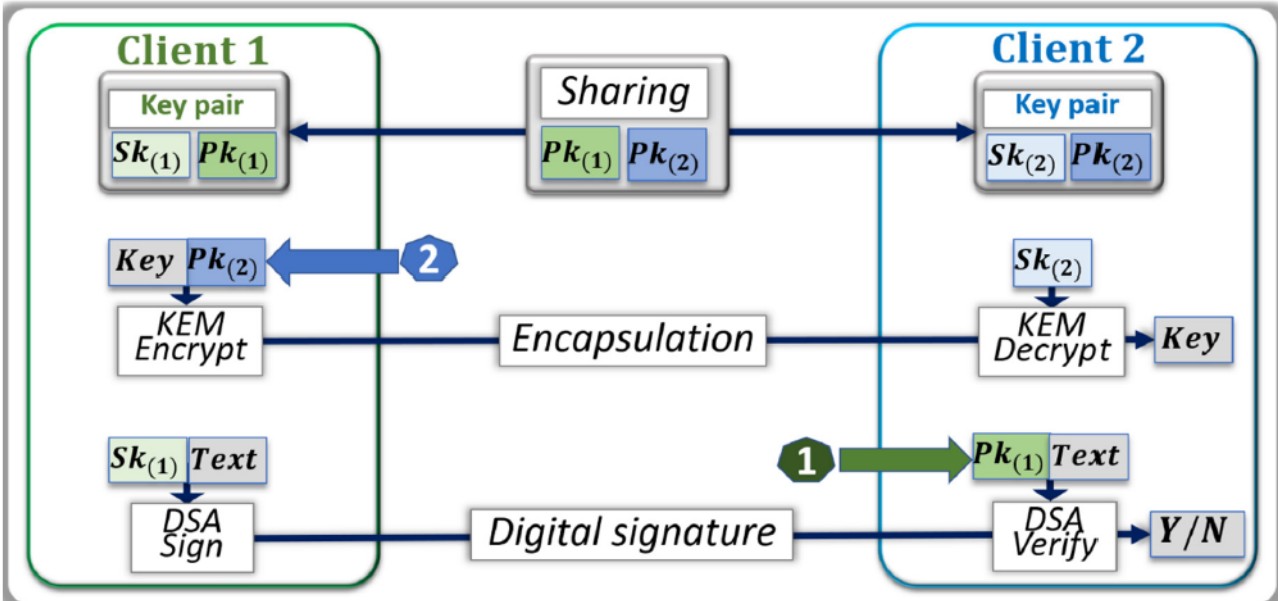

**Figure 4.** Communication protocol between two client devices with shared public keys. Each client device is set with its key pair. For KEM, Client 1 uses the public key of Client 2 to encrypt a shared key; Client 2 retrieves the key with their private key. For DSA, Client 1 signs a message with their private key, Client 2 verifies the message with the public key of Client 1.

Most PKIs rely on certificate authorities (CA) and registration authorities (RA) to offer such an environment of trust and integrity. The architecture is vulnerable to several threats, including loss of identity, man-in-the-middle attacks, and side channel attacks in which the private keys are exposed during KEM, and DSA.

### 3.2. PKI with Network of PUFs

The use of networks of PUFs can mitigate the vulnerabilities of PKIs. PUF technology exploits the variations created during fabrication to differentiate each device from all other devices, acting as a hardware "fingerprint" [27–29]. Solutions based on PUFs embedded in the hardware of each node can mitigate the risk of an opponent reading the keys stored in non-volatile memories. The keys for the PKI can be generated on demand with a one-time use; stealing a key becomes useless as new keys are needed at each transaction. During enrollment cycles, the images of the PUFs are stored in look-up tables in the CA (see Figure 5); enrollment has to be done only once in a secure environment. Handshake protocols [30] can select a portion of the PUFs—and their image is stored in the CA—to extract a data stream that generates the key pairs. The PUFs can be erratic, therefore the generation of cryptographic keys, the focus of this work, is challenging. A single-bit

mismatch in a cryptographic key is not acceptable for most encryption protocols. Therefore, the use of error correcting code (also use the acronym ECC, not to be confused with "elliptic curve cryptography) methods, helper data, and fuzzy extractors can minimize the levels of errors [31–33]. The alternate method is one where the CA has search engines, such as response-based-cryptography (RBC), that can handle the validation of erratic keys [34–37].

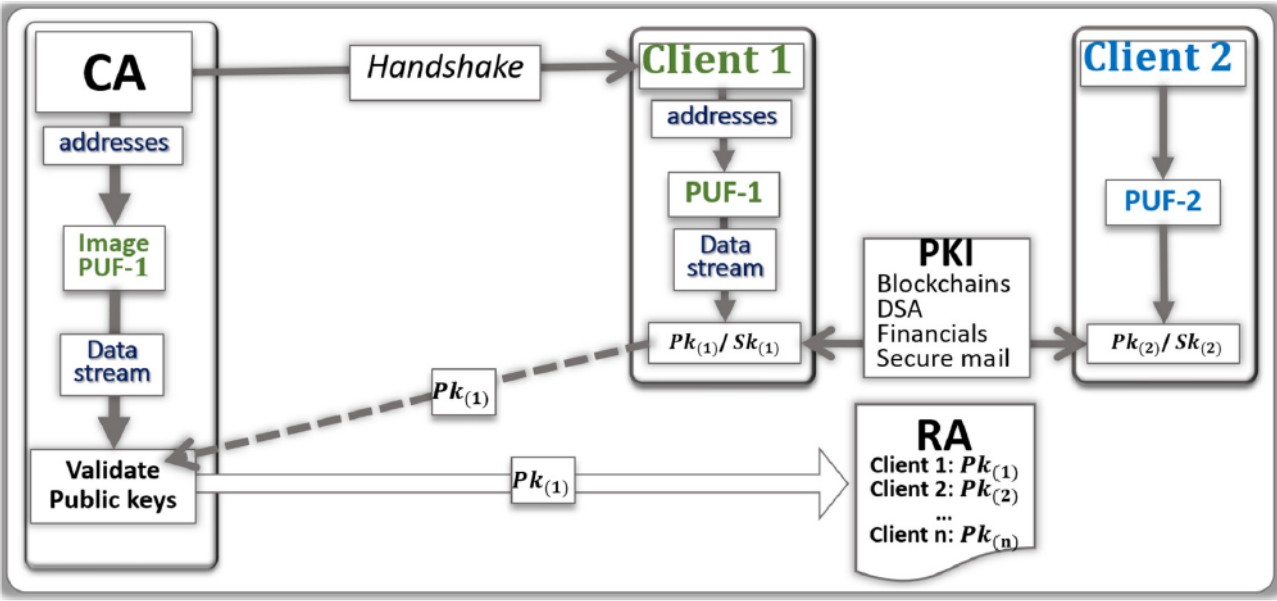

**Figure 5.** Physical unclonable function (PUF)-based public key infrastructure (PKI). The Certificate Authority (CA) verify the validity of the public keys and transfer them to the Register Authority (RA). The key pairs are generated from the PUFs embedded in each client device. The architecture enables cloud-based peer-to-peer secure transactions protected by asymmetrical cryptography.

The RBC engine that validates public keys, shown in Figure 5, generates public/private key pairs until the public key matches the client's provided key. The server searches over a seed (e.g., a 256 bit seed), and uses that seed for key generation. If the generated public key matches the client's public key, then the client is authenticated. If the public keys do not match, then the server flips one bit of the seed at a time (increasing the Hamming distance) until the public keys match. Thus, the search is carried out by generating the public/private key pairs by iterating over the seed and increasing the Hamming distance until the seed is found that matches the client's public key. The search space for a 256 bit key is $2^{256}$ and it would be nearly impossible to authenticate a user in a fixed time without the use of parallel computing. High-performance computing HPC and graphics processing unit (GPU) technologies are valuable to enhance the ability of the CA to validate the public key generated by the client devices. For instance, graphics processing units (GPUs) can be employed to parallelize and accelerate the authentication process. By using a GPU, the server can search over multiple keys in parallel.

### 3.3. Implementation of PQC Algorithms for PKI

The CRYSTALS-Dilithium digital signature algorithm consists of the following procedures: key generation, signing, and verification. These procedures are computationally bound by two operations: multiplication in the polynomial ring noted $\mathbb{Z}_q[X]/(X_n+1)$, and matrix/vector expansion via an extendable output function (XOF). Therefore, any attempt to optimize Dilithium should target these operations. We describe below the literature that focuses on such optimizations.

The operation of polynomial multiplication has a quasi-linear time complexity bound by the Number Theoretic Transform (NTT) implementation, and the operation of expansion via XOF is bound by the SHAKE-128 implementation. Using the AVX2 instruction set,

the matrix and vector expansion is optimized by using a vectorized SHAKE-128 implementation that operates on four sponges that can absorb and squeeze blocks in parallel. Additionally, Ducas et al. [8] use the AVX2 instruction set to optimize the NTT thus speeding up the polynomial ring multiplication by about a factor of two. This optimization is achieved by interleaving the vector multiplications and Montgomery reductions so that parts of the multiplication latencies are hidden.

Nejatollahi et al. [38] outline two different works that optimize the NTT using an Nvidia GPU. The first reports higher throughput polynomial multiplication [39] and the second is a performance evaluation between several versions of the NTT, including iterative NTT, parallel NTT, and CUDA-based FFT (cuFFT) for different polynomial sizes [40]. Strictly algorithmic optimizations of the NTT are presented in other works [41,42]. Longa et al. [41] show that limiting the coefficient length in polynomials to 32 bits yields an efficient modular reduction technique. By employing this new technique in NTT, reduction is only required after multiplication, and significant performance gains are achieved when compared to a baseline implementation. Additionally, the authors use signed integer arithmetic which decreases the number of add operations necessary in both sampling and polynomial multiplication. Greconici et al. [42] use signed integer arithmetic to decrease the number of add operations, which leads to performance gains in several functions including NTT and SHAKE-128. The authors also employ a merging layers technique in NTT that reduces the number of loads and stores by about a factor of two.

The SABER KEM algorithm is similarly computationally bound by polynomial multiplication and hashing functions. As mentioned by D'Anvers et al. [18], since SABER uses power-of-2 moduli, this eliminates the need for rejection sampling and makes modular reduction fast by using bit shift operations. However, one drawback of using power-of-2 moduli is the inability to take advantage of faster NTT multiplication since the moduli are not prime. As described above, Akleylek et al. [40] examines the performance of different multiplication techniques. By implementing a version of cuFFT in a similar fashion for SABER, we may observe a speedup in polynomial multiplication. In addition, SABER is computationally bound by hashing and extendible functions. SABER uses SHA3_256 and SHA3_512 functions for hashing and SHAKE128 as an XOF. Roy et al. [43] demonstrate parallelizing SHAKE128 using AVX2 and batching four operations, thus achieving a 38% increase in throughput for SABER's key generation. Additionally, optimizing the hashing functions and SHAKE128 in a different way, the SABER technical documentation describes replacing the SHA3 functions with SHA2 and replacing SHAKE128 with AES in counter mode [18].

Focusing on three PQC algorithms, SABER, CRYSTALS-Dilithium, and NTRU, a breakdown of the fraction of time spent (as a percentage) in the hashing/XOR and polynomial multiplication components of the algorithms is reported in Table 1. NTRU spends the majority of its time doing polynomial multiplication first, then hashing second [44], but no benchmarks have been calculated thus far. The times spent for the hashing and polynomial multiplication components of CRYSTALS-Dilithium, and SABER are reported as percentages of the total execution time for the key pair generation procedure where the percentages are an average of 10 time trials.

**Table 1.** Breakdown of the fraction of the time hashing and extendable output function XOF compared with the time performing polynomial multiplication. Both SABER and CRYSTALS are constrained by the polynomial multiplication. The light versions considered in the post quantum cryptographic (PQC) implementations of hashing and XOF functions such as SHA3 with SHAKE are extremely effective.

|  | **Hashing and XOF** | **Polynomial Multiplication** | **Reference** |
|---|---|---|---|
| SABER | ~30% | ~60% | Our benchmarks |
| CRYSTALS-Dilithium | ~42% | ~33% | Our benchmarks |
| NTRU | second bottleneck | first bottleneck | [40,41] |

## 4. PUF-Based Key Distribution for PQC

### 4.1. PUF-Based Key Distribution for LWE Lattice Cryptography

The proposed generic protocol to generate public–private key pairs with PUFs for LWE lattice cryptography is shown in Figure 6. The random number generator (a) is used for the generation of seed $a_{(i)}$, which is public information. However, Seed k that is needed for the generation of the private key $Sk_{(i)}$ is generated from the PUF. The outline of a protocol generating a key pair for LWE cryptography is the following:

1. The CA uses a random numbers generator and hash function to be able to point at a set of addresses in the image of the PUF-i.
2. From these addresses, a stream of bits called Seed K' is generated by the CA.
3. The CA communicates to the Client (i), through a handshake, the instructions needed to find the same set of addresses in the PUF.
4. Client (i) uses the PUF to generate the stream of bits called Seed K. The two data streams Seed K and Seed K' are similar, however slightly differ from each other due to natural physical variations and drifts occurring in the PUFs.
5. [If needed, Client (i) applies error correcting codes to reduce the difference between Seed K and Seed K'; the corrected, or partially corrected, data stream is used to generate the vectors $s_{1(i)}$ and $s_{2(i)}$]
6. Client (i) independently uses a random numbers generator (a) to generate a second data stream Seed $a_{(i)}$, which is used for the computation of the matrix $A_{(i)}$.
7. The vector $t_{(i)}$ is computed: $t_{(i)} \leftarrow A_{(i)}\, s_{1(i)} + s_{2(i)}$.
8. The private key $Sk_{(i)}$ is $\{s_{(1(i))};\, s_{2(i)}\}$.
9. The public key $Pk_{(i)}$ is $\{a_{(i)};\, t_{(i)}\}$.
10. Client (i) communicates to the CA, through the network, the public key $Pk_{(i)}$;
11. The CA uses a search engine to verify that $Pk_{(i)}$ is correct. The search engine initiates the validation by generating a public key from Seed $a_{(i)}$ and Seed K' with lattice cryptography codes. If the resulting public key is not $Pk_{(i)}$, an iteration process gradually injects errors into Seed K' and computes the corresponding public keys. The search converges when a match in the resulting public key is found, or when the CA concludes that the public key should be bad.
12. If the validation is positive, the public key $Pk_{(i)}$ is posted online by the RA.

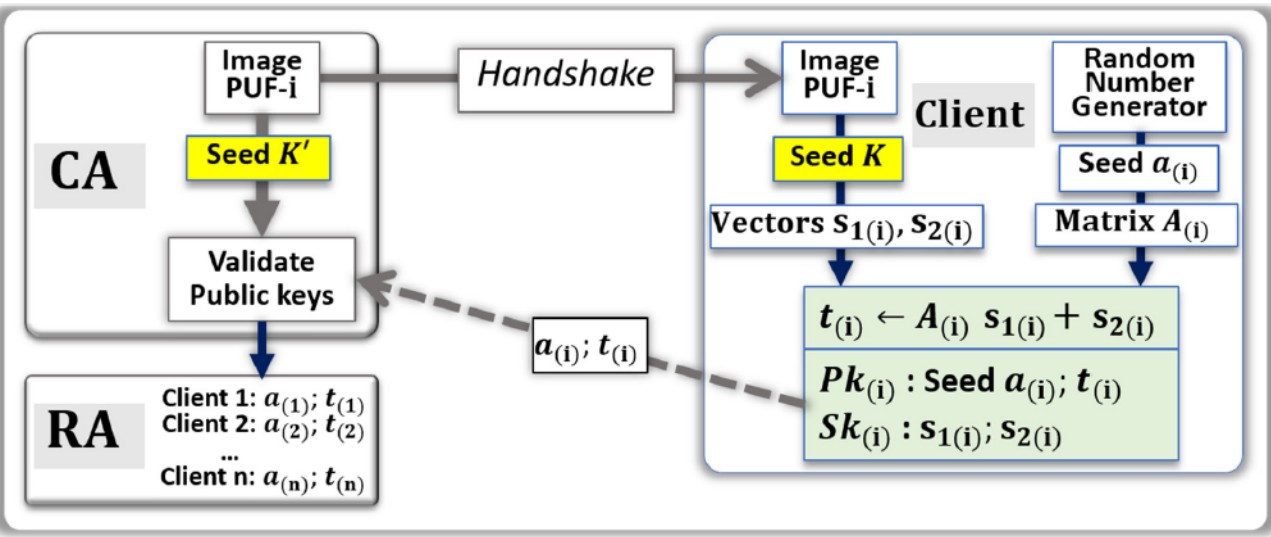

**Figure 6.** PUF-based key pair generation for learning with error (LWE). The private keys, i.e., vectors $S_{1(i)}$ and $S_{2(i)}$, are generated from the seed K that is extracted from the PUF. The matrix $A_{(i)}$ continues to be generated from a random number. The search engine of the CA has access to an image of the PUF, and can independently validate the validity of public key, which is posted by the RA for cloud-based transactions. A new key pair can be generated, and validated by the CA, at each handshake cycle.

This protocol is applicable for single use key pairs that are generated for each transaction. The random number generators of the first step of the protocol can generate new data streams, which point at different portions of the PUFs, thereby triggering the generation of new key pairs. The search engine described above can benefit from noise injection and high-performance computing. The injection of noise in Seed K will make the search too difficult for CA, unless equipped with HPCs, or GPUs. This can preclude hostile CAs from participating.

### 4.2. PKI Architecture with PUF-Based Key Distribution and LWE

The PUF-based key pair generation scheme with LWE cryptography, as presented in the previous section, can be integrated in a PKI securing a network of i clients. Figure 7 shows two client devices communicating directly, either by exchanging secret keys through KEM or DSA. The client devices independently generate the seed $a_{(i)}$, while the PUFs and their images are used for the independent generation of the vectors $s_{1(i)}$ and $s_{2(i)}$. The role of the CA is to check the validity of the vectors $t_{(i)}$, and to transmit both the seeds $a_{(i)}$ the vectors $t_{(i)}$ to the RA, which maintain a ledger with valid public keys. Such an architecture is secured assuming the following conditions:

i. The enrollment process in which the PUFs are characterized to generate their image is accurate and not compromised by the opponent.

ii. The database stored in the CA that contains the image of the PUFs for the i client devices is protected from external and internal attacks.

iii. The PUFs embedded in each client device are reliable, unclonable, and tamper resistant.

iv. The key generation process, KEM, and DSA are protected from side channel analysis.

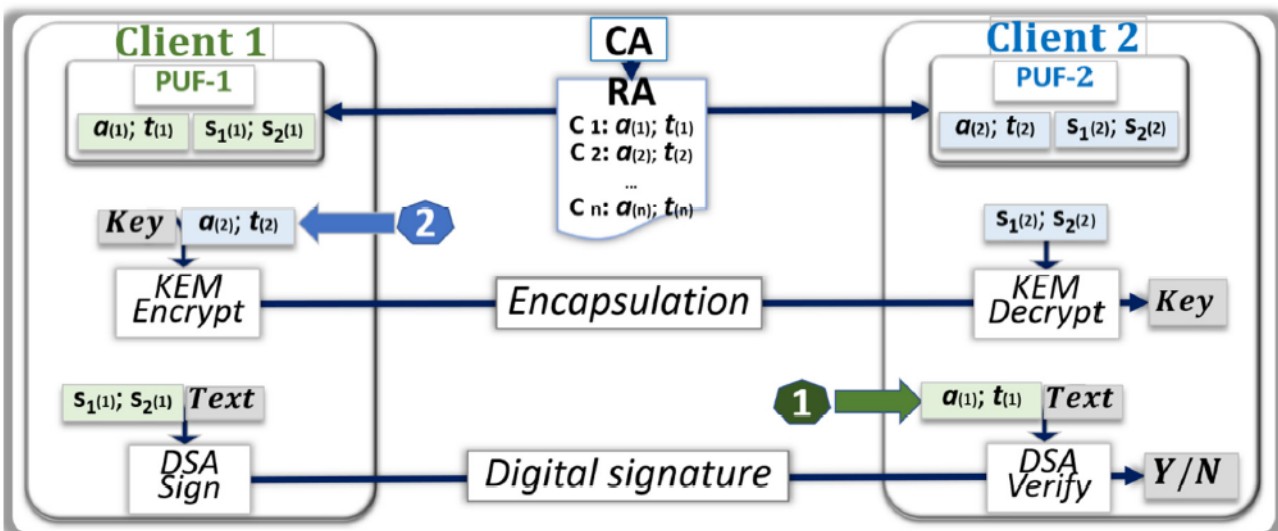

**Figure 7.** PUF-based public key infrastructure (PKI) with LWE cryptography. The use of PUFs in the PKI network does not impact the user experience during peer-to-peer secure communication. The ability to generate one time use key pairs, and authenticate each client device at each transaction, enhances the root of trust. Such PUF based architectures are only valid if the latencies are kept below a few seconds.

As we experimentally verified that the latencies of the key generation process from the PUFs are low enough, such a protocol can be used to change the key pairs after each encryption cycle. Therefore, the potential loss of the secret keys during an encryption/decryption cycle has minimum impact as different keys will be used during the subsequent cycles.

### 4.3. PUF-Based Key Distribution for LWR Lattice Cryptography

There are some similarities between LWE and LWR implementations. The seed k of the PUF is only used to generate one vectors $s_{1(i)}$, while a constant vector $h_{(i)}$ can be generated independently. The public vector $t_{(i)}$ is computed in a similar way: $t_{(i)} \leftarrow A_{(i)} \cdot s_{1(i)} + h_{(i)}$.

### 4.4. PUF-Based Key Distribution for NTRU Lattice Cryptography

The protocol to generate key pairs from PUFs for NTRU cryptography is similar than the one presented above in Section 4.1 for LWE, see Figure 8. We are suggesting a method where the only source of randomness is the PUF, Seed **K**, to compute both the public key $Pk_{(i)}$, and the private key $Sk_{(i)}$. In our implementation, Seed **K** feeds the hash functions SHA-3, and SHAKE, to generate a long stream of bits, then compute the two polynomials $f_{(i)}$ and $g_{(i)}$.

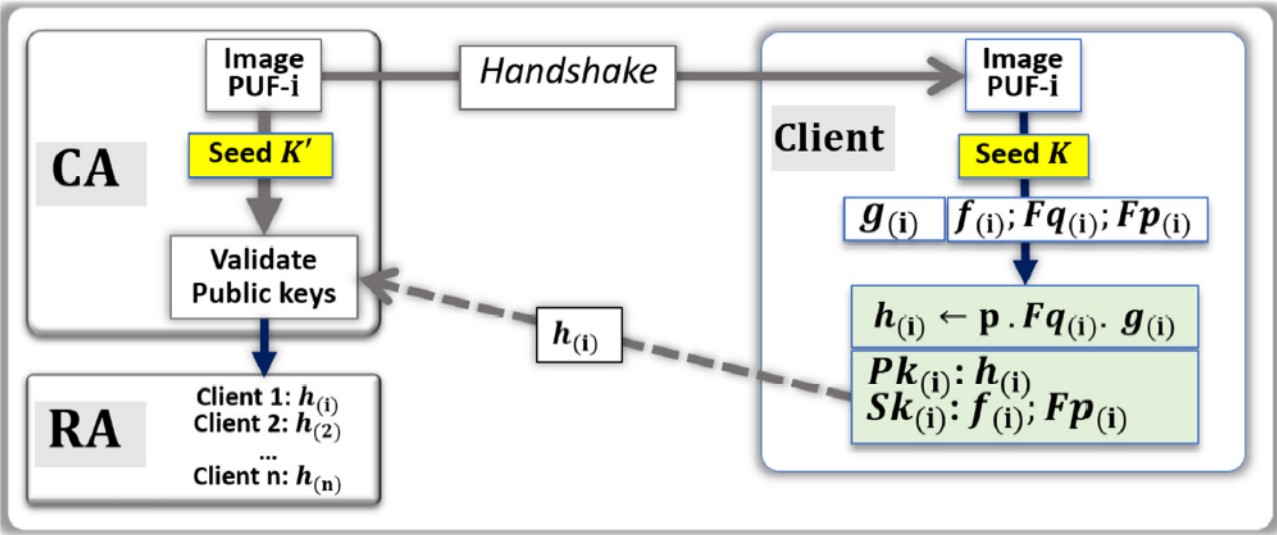

**Figure 8.** PUF-based key pair generation for NTRU. After each handshake, the polynomials $g_{(i)}$ and $f_{(i)}$ are generated from the Seed K that is extracted from the PUF. The public and private key pairs are computed from these two polynomials. The search engine of the CA can independently validate the validity of public key with the image of the PUF.

As previously discussed in Section 2.3, the polynomials $f_{(i)}$ and $g_{(i)}$ are not always usable due to pre-conditions, therefore a scheme to try several possible ways of addressing the PUF has to be developed. One way is to implement a deterministic method that is known by both the client device and the CA, which can have a negative impact on the latencies. We preferred the solution driven by the client device that asks the CA to initiate new handshakes. The summary of the method used to generate the key pairs for NTRU cryptography is the following:

1. The CA uses random numbers to point at a set of addresses in the image of the **PUF-i**.
2. From these addresses, a stream of bits called Seed **K'** is generated by the CA.
3. The CA sends the handshake to the client (i) to find the same addresses.
4. Client (i) uses the PUF to generate Seed **K**.
5. Client (i) applies error correction to Seed **K** and generates the truncated polynomials $f_{(i)}$ and $g_{(i)}$.
6. Computation of $Fp_{(i)}$ and $Fq_{(i)}$ and verify that the pre-conditions are fulfilled.
7. If needed, ask for a new handshake and iterate.
8. The polynomial $h_{(i)}$ is computed: $h_{(i)} \leftarrow p \cdot Fq_{(i)} \cdot g_{(i)}$.
9. The private key $Sk_{(i)}$ is $\{f_{(i)}; Fp_{(i)}\}$.
10. The public key $Pk_{(i)}$ is $h_{(i)}$.
11. Client (i) communicates to the CA, through the network, the public key $h_{(i)}$.

12. The CA uses a search engine to verify that $h_{(i)}$ is correct.
13. If the validation is positive, the public key $h_{(i)}$ is posted online by the RA.

It is important to notice that steps six and seven of the proposed method, "Computation of $Fp(i)$ and $Fq(i)$ and verify that the pre-conditions are fulfilled; if needed ask for a new handshake and iterate", could be handled differently to minimize backward and forward communications cycles between the CA and the client device. One example of implementation is to have a pre-arranged way to modify the seed generated by the PUF and its image. When the CA fails to validate the public key, several pre-arranged modifications of Seed K' will be tested.

### 4.5. PUF-Based Key Distribution for Code-Based Cryptography

An example of a protocol to generate the key pairs with PUFs for code-based cryptography is shown in Figure 9. The overall protocol is similar to the one presented above for lattice cryptography. Much like NTRU, the only source of randomness is Seed **K** that is generated from the PUF to compute the two matrixes $S_{(i)}$ and $P_{(i)}$. The brief outline of the protocol for generating key pairs for code-based cryptography is the following:

1. The CA uses random numbers to point at a set of addresses in the image of the **PUF-i**.
2. From these addresses, a stream of bits called Seed **K'** is generated by the CA.
3. The CA sent the handshake to the client (i) to find the same addresses.
4. Client (i) uses the PUF to generate Seed **K**.
5. Client (i) applies ECC) on Seed **K** and generates the matrixes $S_{(i)}$ and $P_{(i)}$.
6. Computation of $S_{(i)}{}^{-1}$ and $P_{(i)}{}^{-1}$.
7. The public key $Pk_{(i)} = \hat{G}_{(i)}$ is computed with the generator matrix $G$: $\hat{G}_{(i)} \leftarrow S_{(i)} \cdot G \cdot P_{(i)}$.
8. The private key $Sk_{(i)}$ is $\{G; S_{(i)}{}^{-1}, P_{(i)}{}^{-1}\}$.
9. Client (i) communicates to the CA, through the network, the public key $\hat{G}_{(i)}$.
10. The CA uses a search engine to verify that $\hat{G}_{(i)}$ is correct.
11. If the validation is positive, the public key $\hat{G}_{(i)}$ is posted online by the RA.

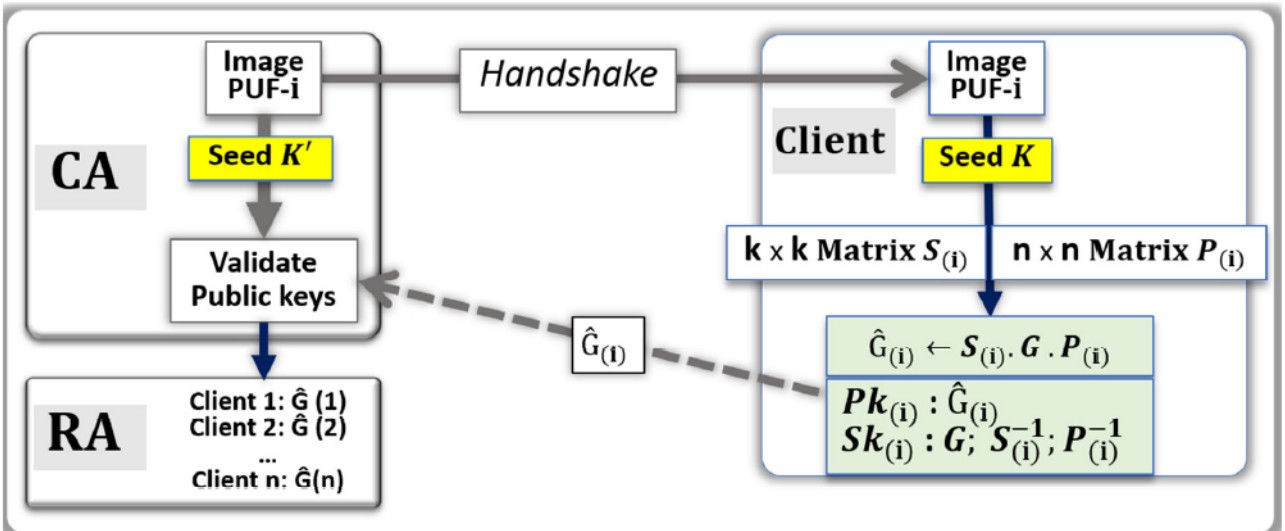

**Figure 9.** PUF-based key pair generation for code-based cryptography. The two matrixes $S_{(i)}$ and $P_{(i)}$ are generated from the Seed K that is extracted from the PUF. As was done in the previously presented PQC schemes, the search engine of the CA has access to an image of the PUF and can independently verify the validity of the public key.

Step six, the computing of inverses, does not work for all matrixes: request for new handshakes, or shared ways to find invertible matrixes and iterate.

## 5. Experimental Evaluation

The purpose of this evaluation is to demonstrate the practicality of the proposed protocols and understand the potential limitations. The replacement of the random number generators by the PUFs follow a similar path for various PQC algorithms; therefore we reduced the scope of this evaluation to LWE (qTESLA, CRYSTALS-Dilithium), and LWR (LightSABER). The generation of the Seed K from the PUF is done using known methods, and the computation of the key pairs is based on the PQC codes made available by NIST, which are also considered known. The unknown in the practicality of the protocol is the sensitivity to bit error rates (BER) of the search engine of the CA for verifying the public keys. The RBC method [34–37] uses the Seed K' as a starting point to generate an initial public key, then iterates by incrementally adding errors, eventually finding the public key computed from Seed K by the client device. At high BER, it is desirable to use cryptographic algorithms that have the ability to generate the key pairs at high throughput.

The RBC itself is an interesting simulation platform for this evaluation, because of the possibility to directly measure the throughput in term of the number of key pairs generated by second. We selected the RBC to experimentally demonstrate that the PQC protocols are fast enough. We designed an experiment to benchmark three algorithms, (qTESLA, CRYSTALS-Dilithium, and LightSABER) with two known cryptosystems (AES and ECC). Each of these cryptosystems have a list of parameter sets as a part of their specifications. We chose parameter sets that were inherently compatible with a 256-bit output from a hypothetical PUF as well as these that were best optimized between performance, size, and security for IoT devices. For these reasons, the parameter sets AES256, ECC Secp256r1, qTESLA-p-I, CRYSTALS-Dilithium 2, and LightSABER were used for the performance comparison. In this analysis, the comparison with ECC is the most relevant one, because the PQC codes under consideration and ECC are similar in their objective to generate public–private keys pairs for PKI. Therefore, ECC and the three PQC algorithms are tested here with their software versions. The comparison with AES was included as a benchmark of excellence; we used the hardware implementation of AES, natively available in Intel processors. One of the objectives of the PQC standardization program driven by the NIST is to encourage private industry to eventually design hardware implementations of the selected codes.

We summarize the parameter sets for each algorithm and our motivation for their selection in Table 2, as shown below.

**Table 2.** Selected cryptosystems, parameter sets, whether the PQC algorithms are NIST round 3 candidates, specialized instructions employed in the implementations, and our motivation for selecting the algorithm and its configuration.

| Algorithm | Parameter Set | NIST Candidate | Instructions | Selection Reason |
|---|---|---|---|---|
| AES | AES256 | N/A | AES-NI, SSE | Benchmark: HW implementation<br>Lacks DSA and KEM capabilities |
| ECC | Secp256r1 | N/A | AVX, SSE | Benchmark: mainstream for PKI<br>Uses 256-bit long keys for DSA |
| qTESLA | p-I | Dropped | AVX, SSE | PQC dropped by NIST: too slow<br>DSA uses relatively small keys |
| CRYSTALS-Dilithium | 2 | Phase 3 | AVX, SSE | Active LWE PQC algorithm<br>One of preferred DSA scheme |
| SABER | LightSABER | Phase 3 | AVX, SSE | Active LWR PQC algorithm<br>One of the preferred KEM scheme |

### 5.1. Experimental Methodology

As of the time of writing, there are few implementations of RBC engines proposed. In this paper, we focus on executing the RBC protocol on a single machine equipped with multi-core CPUs. Our implementations are parallelized using OpenMP. To terminate the search when a thread finds the correct key, we use a flag in shared memory that is

atomically updated. All implementations utilized the same overall structure and key iteration mechanism. We also use AVX instructions in all cryptosystems where applicable; however, further optimizations can be made by taking advantage of AVX2 or other wide vector technologies. The AES256 implementation takes advantage of the AES-NI instruction set, whereas all other cryptosystems tested do not use any additional vectorized instructions except AVX and SSE.

RBC engines targeting purely CPU platforms were only considered for demonstrative purposes. The purpose of this experiment is to compare the relativistic performance between all five chosen cryptosystems. The ease of porting one cryptosystem to another all on the CPU influenced the scope of experiments. Future experimental evaluations exploring GPU focus will require more dedicated, specialized programming for each cryptosystem. The CPU used for the experiments was a $2\times$ Xeon Gold 6132 (Skylake) CPU with 28 total physical cores. Experiments were executed on a dedicated platform. All codes were written in C/C++, compiled using O3 optimization flag.

The experiments were executed by randomly selecting a target thread and using the 256-bit permutation that is the middle of that given thread's workload. This guarantees that each run accurately reflects the average case where execution stops halfway through the 256 choose k search space, for any Hamming distance k. We decided to use this approach to reduce the need for a high number of iterations to reach a statistical central point. Thus, 10 iterations were performed for each cryptosystem, and the median response time was selected from the set of 10 time trials.

The major key performance index (KPI) of the RBC search is key search throughput. Therefore, our performance evaluation uses the "effective key throughput" performance metric. Since the search increases exponentially with Hamming distance, the fraction of time spent initializing/dismantling our procedure will dominate the response time on small workloads (small Hamming distances). Consequently, to measure the effective key throughput, we use a sufficiently large Hamming distance in each algorithm such that we observe peak throughput, indicating that the initialization and dismantling procedures (freeing memory, deconstructing objects, etc.) constitute a negligible fraction of the total response time.

For AES256, the minimum Hamming distance is 4, while for ECC Secp256r1, qTESLA-p-I, CRYSTALS-Dilithium 2, and LightSABER the minimum Hamming distance is 3. Unfortunately, due to the intractable nature of the problem, the single bit error jump from a Hamming distance of 3 to 4 makes it impractical to run a statistically sufficient number of runs for ECC and qTESLA-p-I. For this reason, the AES256 benchmarks ran at a Hamming distance of 4, and the remaining cryptosystems ran at a distance of 3.

*5.2. Evaluation of the Effective Key Throughput*

In this section, we evaluate the effective (peak) key throughput. Figure 10 plots the median of each RBC cryptosystem's effective key throughput on a logarithmic scale. The AES256 implementation, aided by AES-NI, runs several orders of magnitude more efficiently than the public key cryptography variants at $2.17 \times 10^8$ keys per second. ECC Secp256r1 performed the second slowest at $4.77 \times 10^4$ keys per second. The post-quantum algorithms largely performed better than ECC with $1.97 \times 10^5$ and $6.83 \times 10^5$ keys per second for CRYSTALS-Dilithium 2 and LightSABER respectively. qTESLA-p-I was the worst performing PQC and overall cryptosystem out of all five at $2.24 \times 10^4$ keys per second.

To get a better sense of the relativistic scaling, we set ECC Secp256r1's effective key throughput results as the reference point since we are interested in how the PQC algorithms perform when replacing it in future PKI cryptosystems. This is plotted in Figure 11, where now the response variable is displayed in a percentage of the throughout relative to ECC Secp256r1's. Shown here, AES256 is roughly 4550 times more performant than ECC Secp256r1. CRYSTALS-Dilithium 2 is over 4.14 times more efficient than ECC Secp256r1. The most efficient PQC was LightSABER at 14.3 times faster, and the worst overall cryptosystem was qTESLA-p-I at 0.469 times slower.

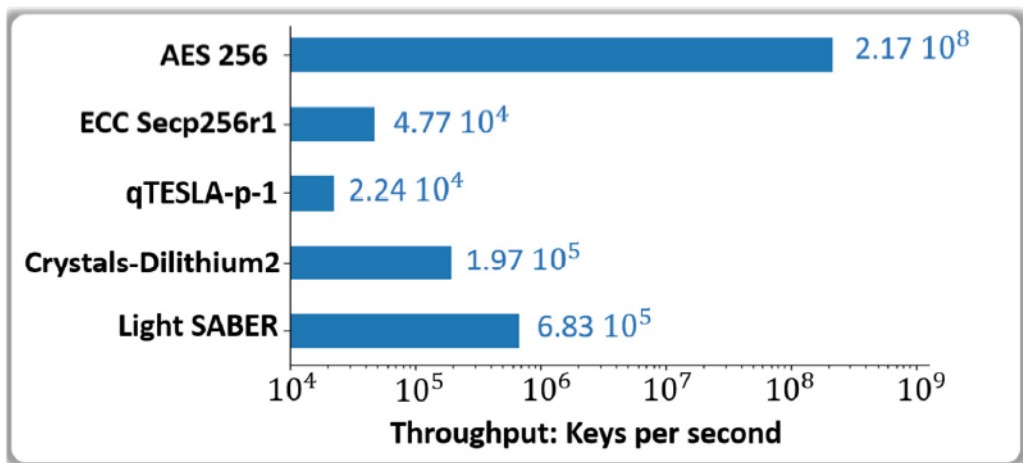

**Figure 10.** Key performance index (KPI) and the effective throughput in keys per second achieved by the response based cryptographic (RBC) search engine powered with AMD Ryzen 9 3900X. Benchmark of the post quantum cryptographic (PQC) algorithms are compared to the reference codes AES 256 and ECC.

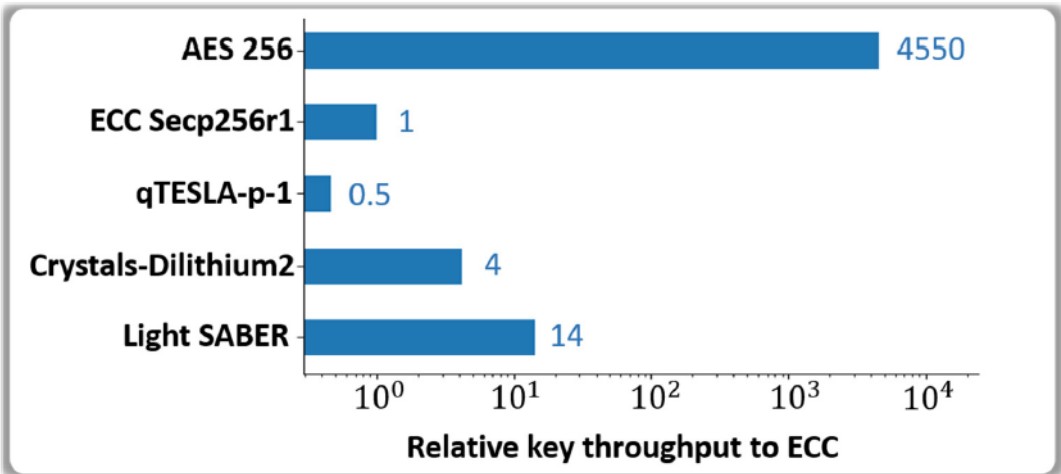

**Figure 11.** Maximum effective throughput relative to the performance of ECC Secp256r1 achieved for each response based cryptographic (RBC) cryptosystem implementation powered by AMD Ryzen 9 3900X. Light SABER performance is approximately 14 times faster than elliptic curve cryptography (ECC).

From these results, we confirm NIST's position that qTESLA is slower than the algorithms selected in round three. Out of what was tested, this leaves CRYSTALS-Dilithium as the strongest candidate for DSA in a PQC environment. For key encapsulation, our results show that SABER is a strong candidate for its relatively fast key generation. Future testing might consider comparing FALCON against CRYSTALS-Dilithium for DSA, and CRYSTALS-Kyber, NTRU, and Classic McEliece against SABER for KEM.

## 6. Conclusions and Future Work

The PQC algorithms under standardization are encouraging and the latencies are reasonable, making the protocols suitable for PKIs securing networks of client devices and IoTs. The generation, distribution, and storage of the public–private key pairs for PQC can be complex because the keys are usually very long. This paper proposes to generate the public–private key pairs by replacing the random number generators with data streams generated from addressable PUFs to get the seeds needed in the PQC algorithms. Unlike the key pairs computed by PQC algorithms, the seeds are relatively short, typically 256-bits long. The use of PUFs as a source of randomness is applicable to all five lattice-based codes

under consideration in the phase III investigation of NIST, and to the code-based Classic McEliece scheme. In order to simultaneously generate key pairs from a server acting as the certificate authority, and the client device with access to its PUF, it is critical to handle the bit error rates (BERs) that are frequent with physical elements. We verified in the experimental section that the RBC can find the erratic seeds by testing an excess of $10^5$ seeds per second with CRYSTALS-Dilithium 2 and LightSABER, which is faster than what we measured with mature algorithms such as the ones with elliptic curves. The experimental evaluation conducted in this work, with the RBC, also lets us conclude that the pre-selection by NIST of CRYSTALS-Dilithium for DSA and SABER for KEM are promising from a performance standpoint. Our results show that the key generation performance is at least comparable to that of ECC. The PQC algorithms under consideration are excellent in an environment targeting PUF-based key exchange. The AES hardware-accelerated AES-NI implementation yields roughly 220 million keys per second throughput on a single machine, which serves a practical real world upper bound for future hardware-accelerated PQC implementations.

In this work we have not yet studied the multivariate-based RAINBOW code, which is also an important scheme under consideration for standardization; we are currently studying ways to use PUFs for key generation. The task needed to deploy PUF-based PQC solutions is not underestimated by the authors of this paper. This will include the use of highly reliable PUFs, and the optimization of the cryptographic protocol pointing simultaneously at the same set of addresses in the PUF, and in the look up table capturing the challenge–response pairs stored in the server. Further optimizing the PUF's protocols and the RBC for PQC algorithms is seen as an opportunity. The use of noises, nonces, errors, and rounding vectors can exploit the stochasticity of PUFs, and the ability to handle erratic streams of the RBC. The PQC algorithms analyzed in this paper can also benefit from the use of distributed memories, high performance computing, and parallel computing, which have the potential to further reduce the latencies of the RBC.

**Author Contributions:** Conceptualization, B.C.; methodology, B.C., M.G., B.Y.; software, M.G., D.G., K.L., S.N., C.P., A.S., J.W.; validation, C.P.; formal analysis, M.G., B.Y., D.G., K.L., S.N., C.P., A.S., B.C.; investigation, M.G., B.Y., D.G., K.L., S.N., C.P., A.S., B.C.; resources, B.C., M.G., B.Y.; data curation, M.G., B.Y., D.G., K.L., S.N., C.P., A.S., B.C.; writing—original draft preparation, B.C., D.G., K.L., S.N., C.P., J.W.; writing—review and editing, B.C., C.P., M.G.; visualization, M.G., B.Y., D.G., K.L., S.N., C.P., A.S., B.C.; supervision, B.C., M.G., B.Y.; project administration, B.C.; funding acquisition, B.C., M.G. All authors have read and agreed to the published version of the manuscript.

**Funding:** This material is based upon the work funded by the Information Directorate under AFRL award number FA8750-19-2-0503.

**Institutional Review Board Statement:** Not applicable.

**Informed Consent Statement:** Not applicable.

**Data Availability Statement:** Not applicable.

**Acknowledgments:** The authors thank the staff, students, and faculty from Northern Arizona University (NAU) in particular, Brandon Salter who is a software engineer in NAU's cybersecurity lab. We also thank the professionals of the Information Directorate of the Air Force Research laboratory (AFRL) of Rome, New York (US), who supported this effort.

**Conflicts of Interest:** The authors declare no conflict of interest. The funders had no role in the design of the study, collection, analyses, interpretation of data, writing of the manuscript, or decision to publish the results.

**Disclaimer:** (a) Contractor acknowledges Government's support in the publication of this paper. This material is partially based upon the work funded by the Information Directorate, under the Air Force Research Laboratory (AFRL); (b) any opinions, findings and conclusions or recommendations expressed in this material are those of the author(s) and do not necessarily reflect the views of AFRL.

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
