# Peer review of "Post Quantum Cryptographic Keys Generated with Physical Unclonable Functions"

_applsci, doi:10.3390/app11062801_

Round 1

Reviewer 1 Report

The introduction can be significantly improved. The introduction of the paper structure can be condensed and the novel contributions of the paper should be articulated.

The quality of figures should be improved, resized, and made self-explainable to ease understanding.

The authors have illustrated the proposed approach without much justification. It is important to relate the proposed techniques to previous studies and provide sufficient motivation before going ahead to give details of the techniques.

The paper is generally hard to read. It is more like a user manual like documentation rather than a scholarly paper. Authors should put sufficient discussion and insights into it.

Author Response

Thank you so much for your feedback. Our response is enclosed in the following pdf file.

Reviewer 2 Report

The authors made a good work, but some points need to be clarified in order the manuscript to get more scientific outlook:

  1. On manuscripts’ title should be written at the end (PUF)
  2. The description of simulation scenario is very poor, and more details need to be provided
  3. The simulation platform that has been chosen by the authors has not been thoroughly justified. The authors should provide an array that provide pros and cons of all available simulation platforms.
  4. There are no KPIs (Key Performance Indexes) on this scenario as well as there no comparisons to other approaches.
  5. On “Conclusion and future work” section the authors need to provide information regarding their contribution to real world
  6. Figures 11 and 12 should be presented in a better way as it is difficult to distinguish the results when is printed in black and white.

Author Response

Thank you very much for your feedback. Our step-by-step response is enclosed in the following pdf file
